# Mirogabalin—A Novel Selective Ligand for the α2δ Calcium Channel Subunit

**DOI:** 10.3390/ph14020112

**Published:** 2021-01-31

**Authors:** Renata Zajączkowska, Joanna Mika, Wojciech Leppert, Magdalena Kocot-Kępska, Małgorzata Malec-Milewska, Jerzy Wordliczek

**Affiliations:** 1Department of Interdisciplinary Intensive Care, Jagiellonian University Medical College, 31-008 Krakow, Poland; renata.zajaczkowska@uj.edu.pl (R.Z.); j.wordliczek@uj.edu.pl (J.W.); 2Department of Pain Pharmacology, Maj Institute of Pharmacology, Polish Academy of Sciences, 31-343 Krakow, Poland; 3Laboratory of Quality of Life Research, Chair and Department of Palliative Medicine, Poznan University of Medical Sciences, 61-701 Poznan, Poland; wojciechleppert@wp.pl; 4Department of Pain Research and Treatment, Jagiellonian University Medical College, 31-008 Krakow, Poland; magdalena.kocot-kepska@uj.edu.pl; 5Department of Anesthesiology and Intensive Care, Medical Center for Postgraduate Education, 01-813 Warsaw, Poland; lmilewski@post.home.pl

**Keywords:** neuropathic pain, α2δ voltage-gated calcium channel subunit ligand, mirogabalin besylate, mechanism of action, pharmacodynamics, pharmacokinetics, clinical indications, drug interactions, adverse effects

## Abstract

The efficacy of neuropathic pain control remains unsatisfactory. Despite the availability of a variety of therapies, a significant proportion of patients suffer from poorly controlled pain of this kind. Consequently, new drugs and treatments are still being sought to remedy the situation. One such new drug is mirogabalin, a selective ligand for the α2δ subunits of voltage-gated calcium channels (VGCC) developed by Sankyo group for the management of neuropathic pain. In 2019 in Japan, mirogabalin was approved for peripheral neuropathic pain following the encouraging results of clinical trials conducted with diabetic peripheral neuropathic pain (DPNP) and postherpetic neuralgia (PHN) patients. The ligand selectivity of mirogabalin for α2δ-1 and α2δ-2 and its slower dissociation rate for α2δ-1 than for α2δ-2 subunits of VGCC may contribute to its strong analgesic effects, wide safety margin, and relatively lower incidence of adverse effects compared to pregabalin and gabapentin. This article discusses the mechanism of action of mirogabalin, presents data on its pharmacodynamics and pharmacokinetics, and reviews the available experimental and clinical studies that have assessed the efficacy and safety of the drug in the treatment of selected neuropathic pain syndromes.

## 1. Introduction

Neuropathic pain is still a challenging problem. It affects hundreds of millions of people worldwide, with an approximate prevalence of 7‒10% in the general population [1]. Neuropathic pain has a negative impact on patients’ physical health and psychological wellbeing, resulting in their poor quality of life. It also poses a major socioeconomic problem for both the individuals concerned and society at large [2].

Despite advances in knowledge leading to a better understanding of mechanisms responsible for neuropathic pain and the efforts of experts worldwide to develop new, more helpful therapies, the efficacy of neuropathic pain treatment remains unsatisfactory. It is estimated that only 50% of treated patients with neuropathic pain achieve 30‒50% pain relief, while the remainder suffer from poorly controlled pain despite the variety of therapies applied [3]. It is therefore unsurprising that new drugs and treatments that could improve the effectiveness of neuropathic pain relief are constantly being sought. One of the new drugs in this armamentarium is mirogabalin.

## 2. General Information

Mirogabalin besylate [(1*R*,5*S*,6*S*)-6-(aminomethyl)-3-ethylbicyclo(3.2.0)hept-3-en-6-yl) acetic acid] is a novel and unique ligand for the α2δ subunits of voltage-gated calcium channels (VGCCs) developed by Daiichi Sankyo for the management of peripheral neuropathic pain (PNP), including diabetic peripheral neuropathic pain (DPNP) and postherpetic neuralgia (PHN) (Figure 1) [4]. In January 2019 in Japan, mirogabalin tablets were approved for the usage of neuropathic pain relief based on trials conducted in DPNP and PHN patients [5]. The name of the mirogabalin product launched on the market is Tarliage, and the available dosage range is 2.5, 5, 10, and 15 mg. Currently, mirogabalin is also in clinical development for PNP in other Asian countries. Research has also been undertaken on the use of mirogabalin for other indications, for example, in fibromyalgia. However, clinical trials of mirogabalin for fibromyalgia-related pain were discontinued in the EU and the USA for the reason that the primary endpoint was not met in 3rd phase [6].

## 3. Mechanisms of Action

Mirogabalin, together with gabapentin and pregabalin, belongs to the so-called gabapentinoids, a group of drugs that affect the α2δ subunits of VGCCs. Given a different classification of calcium channels, experimental studies show that mirogabalin inhibits N-type calcium channel currents in rat dorsal root ganglia (DRG) culture neurons [7]. Mirogabalin has selective and potent binding affinities for human α2δ subunits of VGCCs, which reduce calcium (Ca^2+^) influx and neurotransmission in DRG, inhibiting neurotransmitter release in presynaptic neuron endings [8]. Following inhibited neurotransmitter release (e.g., calcitonin gene-related peptide (CGRP), glutamate, substance P), the hyperexcitability of central nervous system (CNS) neurons decreases, which has a number of pharmacological effects, including analgesic, anxiolytic, and anticonvulsant ones [9]. Four α2δ subunits of VGCCs have been recognized; the α2δ-1 subunit is mainly present in neuronal cells, but the other subunits, α2δ-2, α2δ-3, and α2δ-4, are also present in non-neuronal cells [10]. In addition, the α2δ-1 subunit is also present in skeletal, cardiac, and smooth muscles, whereas α2δ-4 is present in retinal neurons [10].

Several studies have shown that α2δ subunits of VGCC are significant for the correct trafficking and physiological function of the calcium channels [11]. It was described that α2δ-1 subunits of VGCC play a key role during neuropathic pain development, which results from injury to sensory nerves [12]. It is also known that α2δ-1 subunits of VGCC are strongly up-regulated in somatosensory neurons following nerve damage [13]. Moreover, α2δ-1 and α2δ-2 subunits of VGCC were identified as the therapeutic target for the gabapentinoids [8,14,15].

Behavioral, electrophysiological, and neurochemical studies using transgenic mice have shown that the α2δ-1 subunit of VGCCs contributes to analgesic effects of gabapentinoids [16,17], whereas the α2δ-2 contributes to CNS side effects [18]. Compared to pregabalin, mirogabalin shows stronger binding affinities for the α2δ-1 and α2δ-2 subunits and a slower dissociation rate for the α2δ-1 than α2δ-2 subunits [19]. It is proposed that the mirogabalin’s selectivity for α2δ-1 and α2δ-2 and its slower dissociation rate for the α2δ-1 than α2δ-2 subunits of VGCC might contribute to its higher analgesic efficacy, wider safety margin, and relatively lower incidence of CNS adverse effects (AEs) compared to pregabalin and gabapentin and, in effect, to different clinical outcomes [19].

There is another mechanism that may affect effectiveness of α2δ-1 subunits ligands. Recent studies indicate that in neuropathic pain caused by chemotherapeutic agents and peripheral nerve injury the α2δ-1 subunit of VGCCs can form a complex with *N*-methyl-d-aspartate receptors (NMDARs) through its C-terminus and regulate their synaptic trafficking and activity in the brain and DRG [20]. It is also suggested that gabapentinoids, by targeting α2δ-1 subunits of VGCCs which are combined with NMDARs, can beneficially relieve symptoms of neuropathic pain [20]. In experimental models of neuropathic pain, pretreatment with *N*-methyl-d-aspartate (NMDA) antagonist (MK801) significantly increased the antinociceptive effect of pregabalin what suggests that gabapentinoid’s antinociception is also mediated through NMDA receptor’s involvement [21]. A summary of mechanisms of action and therapeutic effects of mirogabalin is shown in Figure 2.

## 4. Pharmacodynamics

It is known from the in vitro study that mirogabalin bind selectively and with a high affinity to the α2δ-1/-2 subunits of VGCCs. Its dissociation constants (K_d_) were determined as 13.5 mmol/L for the α2δ-1 subunit and 22.7 nmol/L for the α2δ-2 subunit [19]. The binding of the α2δ-1 subunit of VGCCs by mirogabalin is more potent than the binding by pregabalin because the K_d_ in human α2δ-1 subunit of VGCCs for pregabalin is 62.5 nmol/L [19]. Compared to pregabalin, mirogabalin binds to the α2δ-1 and α2δ-2 subunits of VGCCs for a longer time. This applied mainly to the α2δ-1 subunit of VGCCs: the dissociation half-life from the α2δ-1 subunit is 11.1 h, and from the α2δ-2 subunit, it is 2.4 h for mirogabalin vs. 1.4 h from both α2δ-1 and α2δ-2 subunits for pregabalin [19]. These differences explain the diverse efficacy and safety profiles of mirogabalin compared to other gabapentinoids because the α2δ-1 subunits of VGCCs have been linked with gabapentinoids analgesic properties [9], whereas the α2δ-2 subunits of VGCCs with the adverse CNS effects [22]. According to experimental data, mirogabalin’s safety profile is superior to pregabalin’s, which is evidenced by the comparison of dosages that produce 50% of the maximum adverse effect and 50% of maximum analgesia, respectively [19].

## 5. Pharmacokinetics

Mirogabalin was tested in healthy volunteers at doses ranging from 3 to 75 mg [23]. After oral administration, mirogabalin is quickly absorbed, with a mean time to maximum plasma concentration (T_max_) of 0.5–1.5 h after single or repeated doses [5]. The plasma maximum concentration (C_max_) and concentration–time curve (AUC) increased in a dose-dependent manner, whereas the steady-state plasma concentration was achieved by day 3 [23]. In the fed state, T_max_ is delayed by 0.5 h after taking a 15 mg dose of mirogabalin, and C_max_ is reduced by approximately 18%, but it does not influence its general exposure to any clinically relevant degree [23]. Mirogabalin has a low plasma protein binding of approximately 25%. In patients with severe liver damage, the mean plasma protein binding of mirogabalin is reduced only marginally (to approx. 22.1%) [24]. It was shown, that after administration mirogabalin is quickly converted into its free form in which A200-700 is the main active circulating isoform [24]. The main cytochrome P450 isoenzymes are not induced or inhibited by mirogabalin [9]. The drug is cleared mainly unchanged (61–72%) via renal excretion by filtration and active secretion, however a slight fraction (13–20%) is metabolized by hepatic uridine 5′-diphospho-glucuronosyltransferase isoforms [24]. The mean elimination half-life of mirogabalin observed in clinical trials was 2‒3 h [25], 3.5 h [26], and 3–4.9 h [23]. Ninety-nine percent of mirogabalin is excreted through the kidneys, with only 1% of the dose excreted in the feces. Mirogabalin is excreted renally via both tubular secretion and glomerular filtration. Urinary metabolites of mirogabalin contain lactams and an *N*-glucuronide conjugate [5]. In patients with renal impairment, it is necessary to modify the dosage of the drug. A dose adjustment of 50% is needed in people with moderate renal impairment (creatinine clearance (CrCl) 30–50 mL/min/1.73 m^2^) and of 75% in people with severe renal impairment (CrCl < 30 mL/min/1.73 m^2^) [27]. It does not appear necessary to alter the dosage of mirogabalin in persons with mild or moderate hepatic impairment relative to the regimen for subjects with normal hepatic function. Generally, a single dose of mirogabalin (15 mg) is well tolerated by patients with moderate and mild hepatic impairment [24].

## 6. Efficacy

### 6.1. Experimental Trials

In rodent neuropathic pain models, such as streptozotocin-induced (STZ-induced) diabetes and partial sciatic nerve ligation, mirogabalin revealed longer-lasting and more potent analgesia than pregabalin [19]. Unlike pregabalin, in diabetic neuropathy in rats after repeated mirogabalin administration the analgesic effects were more pronounced [19]. In pharmacological evaluations, the safety indices of mirogabalin were superior to those of pregabalin. The authors concluded that these properties of mirogabalin and the differences from pregabalin may be associated with the former’s unique binding characteristics [18]. Moreover, mirogabalin was found to have selective binding affinities for α2δ-1 subunit of VGCCs at doses of 10 and 30 mg/kg, with a peak analgesic effect at 4 h, which remained effective for 6–8 h [19].

The analgesic potential of mirogabalin was also studied in two animal models of fibromyalgia: unilateral intramuscular acidic saline injection (Sluka model) and intermittent cold stress (ICS model). In both models, long-lasting increases in the pain response score were observed. In those studies, mirogabalin administered per os at doses of 1, 3 or 10 mg/kg showed a significant, long-lasting (even up to 8 h), dose-dependent decrease in the total pain score in response to the von Frey test stimulus in those animal models of fibromyalgia [28]. Therefore, the authors proposed mirogabalin as potential drug for pain relief in patients with fibromyalgia [28].

In 2018, Domon et al. [29] investigated the influence of mirogabalin on nociceptive transmission in a rat model of spinal cord injury for neuropathic pain. The spinal cord injury model was established by acute compression of the spinal cord at the T6/7 level with a microvascular clip in male rats. Twenty-eight days after spinal cord injury, when animals received mirogabalin, they had lower tactile hypersensitivity as measured by the von Frey test. A single oral administration of mirogabalin at doses of 2.5, 5, or 10 mg/kg, significantly increased the paw withdrawal threshold. The effects of mirogabalin were still significant 8 h after administration. The authors proved that mirogabalin showed potent and long-lasting analgesic effects in a rat model of spinal cord injury and suggested that this drug may provide effective pain relief in patients with neuropathic pain caused by CNS injury [29].

In another experimental study, Murosawa et al. [30] investigated the anxiolytic effects of mirogabalin in an experimental model of neuropathy. After chronic constriction injury of the sciatic nerve (CCI model) in rats, hypersensitivity developed, which was measured by von Frey test. Anxiety-related behavior was evaluated using the elevated plus maze test. A single oral mirogabalin administration dose-dependently (3–10 mg/kg) alleviated the abovementioned tactile hypersensitivity and anxiety-related behaviors. The authors concluded that mirogabalin may provide effective pain and anxiety relief in patients with neuropathic pain [30]. Importantly, mirogabalin also showed an anxiolytic effect in the fibromyalgia rat model (Sluka model) [31,32]. In other experimental studies, mirogabalin showed a protective potential over multiple brain functions against repeated restraint stress in mice. To estimate learning function, anxiety levels, and hippocampal neuronal activities, Y-maze, elevated-plus maze and c-Fos immunohistochemistry have been applied. It was shown that pre-emptively administered mirogabalin prevented memory dysfunction, anxiety-like behavior, an abnormal defecation score, and increased hippocampal c-Fos expression, which the authors believe may be mediated by inhibition of hippocampal neuron hyperactivation [33].

### 6.2. Clinical Trials

#### 6.2.1. Diabetic Peripheral Neuropathic Pain

To date, several studies have been conducted in patients with diabetic peripheral neuropathic pain (DPNP) to evaluate the mirogabalin safety and efficacy. Vinik et al. [34] conducted a randomized, double-blind, and multicenter phase 2 clinical study. Adult patients with DPNP with a duration of above 6 months were randomized to receive placebo, mirogabalin (5, 10, 15, 20, 30 mg daily) and pregabalin (300 mg daily) for 5 weeks and the pain severity was measured. A meaningful effect was defined as at least a one-point change in the daily mean pain scores (numerical rating scale, NRS 0–10). Mirogabalin 30 mg/d decreased them by one point or more, whereas 20 mg/d led to statistically significant pain relief, which, however, was not considered clinically relevant. Significant differences in daily mean pain scores were noted between pregabalin 300 mg/d and mirogabalin 30 mg/d showing superiority of the latter. Median times to meaningful pain relief in comparison with placebo administered for 36 d, were 30, 20, and 16 d in patients administered mirogabalin 10, 20, and 30 mg/d, respectively. In the present study, the researchers assessed the safety profile of mirogabalin using clinical laboratory tests, electrocardiograms and adverse event data. The results strongly suggests that the mirogabalin was well tolerated. The most frequent adverse effects were mild to moderate and included headache (6.1%), somnolence (6.1%), and dizziness (9.4%) [34].

In 2019, Baba et al. [35] made an open-label extension examination in Asian patients. These findings provide strong evidence of the efficacy of long-term mirogabalin administration for pain relief in patients with diabetic neuropathy. Participants in the study were initiated on mirogabalin 5 mg twice daily, which was subsequently increased to 10/15 mg twice daily. Of the 214 patients who entered the study, 172 (80.4%) completed it and 149 of them received a high dose of mirogabalin (15 mg twice daily). The visual analogue scale (VAS) and McGill Pain Questionnaire (short-form) sensory, affective, total pain, and present pain intensity scores were gradually reduced until week 52. Mirogabalin was well tolerated, and no significant safety concerns were identified in the long-term flexible dosing regimen. In general, the incidence of treatment-emergent adverse events was 91.1%, most of which were mildly or moderately severe. They included nasopharyngitis (27.1%), diabetic retinopathy (11.7%), peripheral edema (11.2%), somnolence (9.3%), diarrhea (8.4%), increased weight (7.9%), and dizziness (7.5%). Treatment-emergent adverse events leading to the discontinuation of treatment occurred in 13.1% of cases [35].

In a recently published review, which included three randomized controlled trials with a total of 1732 patients with DPNP, mirogabalin was superior to the placebo and pregabalin in decreasing the average daily pain scores. The most often side effect observed in mirogabalin group were dizziness, increased weight, peripheral oedema, and somnolence [36].

The studies completed to date suggest that mirogabalin is a promising treatment modality for diabetic neuropathy-induced pain. However, further research is needed to assess the efficacy and safety of mirogabalin in this context.

#### 6.2.2. Postherpetic Neuralgia

In 2019, Kato et al. [37] examined the efficacy of mirogabalin in the treatment of postherpetic neuralgia (PHN) (NCT02318719). In this multicenter, double-blind, phase 3 study, Asian patients aged 20 and above suffering from postherpetic neuralgia were assigned at random to the placebo or mirogabalin groups, with the latter being given 15, 20, or 30 mg of the drug daily for up to 14 weeks. Of the 763 randomized patients included in the analysis, 671 (87.7%) completed the study. At the end of the study, the difference in average daily pain scores in the mirogabalin 15, 20, and 30 mg/d groups was −0.41, −0.47, and −0.77, respectively. The findings in all the mirogabalin groups showed statistical significance. The proportion of patients reporting a ≥30% reduction in neuropathic pain was significantly higher compared to the placebo in all three dosing regimens of mirogabalin. In PHN patients the drug was shown to be superior to the placebo, but its reported efficacy in pain relief studies may be lower because patients with high pain scores (≥9/10) were excluded from the study. The most common adverse events were mild or moderate in severity, namely, somnolence, weight increase, nasopharyngitis, dizziness, and peripheral edema. All the mirogabalin dosage regimens significantly improved sleep quality; moreover, considerably more patients declared their overall health as “much improved or better” (in the 15 mg/d mirogabalin group) after 14 weeks of treatment as compared to the placebo group [37].

The next open-label extension studying the phase 3 long-term administration was conducted during one year. In patients with PHN the analgesic mirogabalin beneficial effects were observed over 52 weeks. The mean pain intensity measured by VAS was 29–35 mm over weeks 12–52 vs. 44 mm at baseline [5]. Due to the limited amount of data, further studies are needed to evaluate the efficacy and safety of mirogabalin in the management of PHN-related pain, although the results currently available are encouraging.

#### 6.2.3. Fibromyalgia

In 2019, Arnold et al. published the results of multicenter, double-blind phase 3 studies (NCT02146430, *n* = 1293; NCT02187159, *n* = 1270; NCT02187471, *n* = 1301) provided for 13 weeks with patients with fibromyalgia randomized to placebo, pregabalin (150 mg, twice daily), mirogabalin (15 mg, once daily), and mirogabalin (15 mg, twice daily). Patients receiving mirogabalin did not achieve significant average daily pain scores relief. In contrast, pregabalin reduced average daily pain scores for baseline vs. placebo (NCT02187159; NCT02187471). Moreover, the authors also proved the long-term (52 weeks) safety of mirogabalin (NCT02234583), but the primary endpoint of significant pain reduction was not achieved [38].

#### 6.2.4. Other Research

In 2020, Tetsunaga et al. [39] investigated the outcomes of mirogabalin administration in patients with peripheral neuropathic pain who terminated treatment with pregabalin due to the lack of efficacy or adverse events. Depending on pain symptoms and age of patient, the dose of mirogabalin was reduced or increased as required to between 2.5 and 15 mg twice daily. In patients with impaired renal function, the starting dose was decreased to 5 mg/d. In total, the mirogabalin was administered to 187 patients with the following problems: carpal tunnel syndrome (9 patients), lumbar disc herniation (10 patients), cervical spondylotic myelopathy (33 patients), and lumbar canal stenosis (134 patients). Patients received mirogabalin 10 mg/d orally for the first week. It was observed that after 1 week of mirogabalin treatment, the NRS scores were diminished. After 8 weeks, in 113 patients (69.3%) the NRS scores improved by ≥30% from baseline. Due to adverse events, 24 patients (12.8%) stopped mirogabalin management. The most frequent adverse events of mirogabalin included somnolence (26.7%), dizziness (12.3%), peripheral edema (5.9%), and weight gain (0.5%) [21]. This study shows that mirogabalin may be a good therapeutic option for patients suffering from neuropathic pain when pregabalin is ineffective.

Central pain syndromes represent a form of neuropathic pain that is associated with injury of the CNS, most often after a stroke or a traumatic injury. This type of pain is very difficult to treat. Importantly, a randomized, placebo-controlled, double-blind trial (NCT03901352) is ongoing in patients to examine the safety and efficacy of mirogabalin in case of pain related to brain and spinal cord injury [40].

## 7. Safety Profile/Adverse Effects

In 2018, Brown et al. [23] analyzed the pharmacokinetics and pharmacodynamics of mirogabalin in healthy subjects (males and females) in order to assess its safety and tolerability in fasted and fed states with a view to determining its most appropriate dosage. The drug proved to be well tolerated in both states. The most common side effects that lead to treatment discontinuation were somnolence and dizziness, they were found to be dose-dependent [23]. In 2018, a randomized, double-blind study was carried out by Jansen et al. [25] in order to evaluate the effects of single (10‒40 mg) and repeated (10‒15 mg 2× daily) mirogabalin doses on Chinese, Japanese, Korean, and Caucasian patients. The medication was quickly absorbed within 1 h and eliminated within 2‒3 h. In Asian and Caucasian patients, mirogabalin administered in doses of up to 15 mg 2× daily for 7 d was found to be sufficiently safe and tolerable. The drug did not accumulate on retreatment. Its most commonly reported adverse events included somnolence, headache, and dizziness [25].

In 2016, nonlinear mixed-effects analyses were conducted by Hutmacher et al. [41] with a view to estimating the conversion ratio from mirogabalin to pregabalin, the possibility of significant pain relief using both drugs, and the incidence of adverse events. The researchers found that mirogabalin was ca. 17 times more effective than pregabalin. The incidence of the most common side effects (dizziness/somnolence) decreased over time and was reduced by titration. The authors concluded that mirogabalin administered twice daily is less likely to cause dizziness than once daily doses [41].

In 2019, Baba et al.’s [35] study assessed the long-term (52 weeks) safety and efficacy profile of mirogabalin in 214 Asian patients suffering from diabetic polyneuropathy. Eighty percent (172 patients) completed the study. The most common side effects included nasopharyngitis (27.1%), diabetic retinopathy (11.7%), peripheral oedema (11.2%), somnolence (9.3%), diarrhea (8.4%), weight gain (7.9%), and dizziness (7.5%). Most of them (i.e., somnolence, weight gain, dizziness, and peripheral oedema) were described as mild to moderate and resolved without treatment [35].

Gabapentin and pregabalin bind to the α2δ-1 and α2δ-2 subunits nonselectively, and produce unwanted side effects in the CNS. The α2δ-2 subunit of VGCCs is principally located in the cerebellum [10], and therefore its activation may cause dizziness, somnolence, headache, cerebellar ataxia, peripheral edema, fatigue, and blurred vision. These CNS-related adverse reactions were reported for both drugs (29.1% with gabapentin and 35.2% with pregabalin) [42,43]. While mirogabalin produces a lower level of CNS-specific adverse drug reactions, due to a low affinity to, and rapid dissociation from the α2δ-2 subunits of VGCCs in the cerebellum [44]. Mirogabalin shares the three main adverse reactions of gabapentinoids, namely, dizziness, somnolence, and headache [42]. However, according to Vinik et al. [34], in mirogabalin-treated patients, the adverse effects were mild to moderate and occurred with frequency: dizziness—9.4%, somnolence —6.1%, and headache—6.1%.

The pharmacokinetics of mirogabalin in healthy elderly patients (55–75 y) [45] is comparable to that in the nonelderly [5]; however, due to the fact that elderly patients may suffer from decreased renal function, they are more at risk of falls caused by somnolence or dizzy spells and were listed among the potential adverse events. For this reason, mirogabalin should be used with caution in this patient group. In summary, mirogabalin appears to be a drug with a good safety profile. Most of the side effects are of mild or moderate intensity; they occur more frequently with a rapid dose increase and once a day dosing, and their intensity gradually decreases with the duration of the therapy, relatively rarely leading to the discontinuation of the therapy.

## 8. Risk of Addiction

In agreement with US Food and Drug Administration guidelines, in double-blind, randomized, placebo- and active-controlled crossover studies in recreational polydrug users, the abuse potential of mirogabalin (range of doses 15–105 mg) was estimated with comparison to placebo, diazepam (15, 30 mg), and pregabalin (200, 450 mg) [26]. The primary endpoint adopted was the maximum observed effect (Emax) in the drug liking visual analog scale. At therapeutic doses, the Emax of mirogabalin did not differ from placebo and was lower than either pregabalin or diazepam. Furthermore, in supratherapeutic doses (≥4×) it had a higher Emax than the placebo, which, however, was still lower than pregabalin. These findings suggest that mirogabalin has a limited potential for abuse in comparison with diazepam or pregabalin, but more importantly, it was well tolerated by persons with a history of multidrug use for recreational purposes [26]. Due to the growing number of fatalities attributed to pregabalin and gabapentin associated with their misuse along with either opioids or multidrug abuse [46], mirogabalin may represent a safer alternative therapeutic option with fewer sedative effects.

## 9. Drug Interactions

Mirogabalin does not induce or inhibit cytochrome P450 isoenzymes. Coadministration of mirogabalin with cimetidine or probenecid may raise the mirogabalin plasma concentration [5], although this increase in exposure may not be clinically relevant [5,47]. In 2018, Tachibana et al. [47] studied drug-to-drug interaction between mirogabalin and probenecid or cimetidine. Probenecid and cimetidine are known to inhibit renal clearance, probenecid also diminished metabolic clearance. The increase in mirogabalin exposure when coadministered with cimetidine was like that observed with mild renal impairment, but in these patients mirogabalin dose adjustments are not necessary. Similarly, the effect of probenecid on mirogabalin exposure is not significant [47].

Importantly, if mirogabalin is taken with alcohol or lorazepam, the depressive effects on the CSN may be potentiated [5,48]. Randomized, double-blind, placebo-controlled, 4-period drug-drug interaction studies were provided in healthy patients to estimate the pharmacodynamic and pharmacokinetic interactions between mirogabalin and other drugs. Mirogabalin was administered alone or with single-dose lorazepam, zolpidem, tramadol, and ethanol. For 48 h after drug administration the safety assessment and samples for pharmacokinetic parameters were analyzed. Mirogabalin coadministered with zolpidem increased the incidence of somnolence, with tramadol and ethanol—headache and nausea, but with lorazepam and ethanol—dizziness and somnolence. Patients should be informed about the potential adverse effects when mirogabalin is coadministered with other drugs and substances, acting on the CNS [48].

A phase 1 study was conducted in healthy subjects to evaluate the possibility of drug-drug interactions (DDIs) between single doses of metformin and mirogabalin. Both compounds share a common drug transporter for renal elimination and may be coadministered in DPNP patients. Coadministration of mirogabalin and metformin was well tolerated in healthy subjects with no evidence of any drug-drug interaction [49].

## 10. Renal and Hepatic Impairment

### 10.1. Renal Impairment

Mirogabalin is readily absorbed into systemic circulation and is eliminated mostly by the kidneys. For this reason, reduced mirogabalin dosage is recommended in patients with moderate-to-severe impairment of renal function. Kato et al. [50] completed a multicenter, open-label study in a Japanese population with a view to determining the effect of a single mirogabalin injection (5 mg) on its pharmacokinetics and safety in patients with varying degrees of such impairment. They found that renal clearance (CLr), total body clearance (CL/F), and the cumulative mirogabalin dose-dependent percentages excreted into urine were reduced proportionally to the severity of the impairment. Based on these findings, the authors suggested that mirogabalin should be carefully titrated in patients with moderate-to-severe renal impairment and those with end-stage renal disease [50].

In 2016, Yin et al. [27] published the findings of a study assessing the impact of renal dysfunction on the efficacy and safety of mirogabalin. A population pharmacokinetic model was established in order to determine the plasma concentrations of mirogabalin and lactam metabolites depending on the severity of kidney damage. The total clearance of a single oral dose of mirogabalin (5 mg) decreased by 25%, 54%, and 76% in persons with mild, moderate, and severe renal impairment, respectively. These data suggest that a 75% reduction of the drug dose is necessary in people with severe (CrCl < 30 mL/min/1.73 m^2^) and 50% reduction in people with moderate (CrCl 30-50 mL/min/1.73 m^2^) impairment of renal function. Nevertheless, no dosage adjustment is necessary in patients with mild renal impairment (CrCl 50–80 mL/min/1.73 m^2^) [27]. A phase 3 trial assessing the efficacy and safety of mirogabalin in kidney disease (ClinicalTrials.gov Identifier NCT02496884) has been completed, but its findings have not yet been published [51].

### 10.2. Hepatic Impairment

In 2018, Duchin et al. [24] conducted an open-label study to determine how mirogabalin pharmacokinetics changes in patients with mild-to-moderate hepatic impairment. The authors analyzed the time to maximum concentration and the area under the concentration time curve until the last quantifiable concentration of the active free form (A200-700) and inactive lactam metabolite (A204-4455). There were no moderate/severe adverse effects or discontinuations of treatment following mirogabalin administration (15 mg, daily) in patients with mild-to-moderate hepatic impairment. Two of the eight patients reported somnolence, but both rapidly recovered. This study provides evidence that it is not necessary to modify the dosage of mirogabalin in patients with mild-to-moderate hepatic impairment [24]. On account of the promising results of the study and the small number of patients, similar lines of inquiry should be pursued in the near future.

## 11. Conclusions

Due to the unsatisfactory efficacy of neuropathic pain treatment, new drugs and methods that could improve neuropathic pain are being sought. One of the new drugs in neuropathic pain treatment is mirogabalin, a selective ligand for the α2δ-1 and α2δ-2 subunits of the VGCC, developed by Daiichi Sankyo for the treatment of PNP and approved in 2019 in Japan following trials conducted in patients with DPNP and PHN. The currently available studies demonstrate the high analgesic efficacy of mirogabalin in relieving pain associated with DPNP and PHN, its wide safety margin, and the relatively lower incidence of adverse effects compared to those of pregabalin and gabapentin. Development of mirogabalin for fibromyalgia pain was discontinued in phase 3 trials in the USA and EU. Further research is necessary to determine the place of mirogabalin in the treatment of neuropathic pain of different etiology, including central pain, to validate its long-term analgesic efficacy and safety, as well as the compatibility of mirogabalin with other neuropathic pain treatment protocols.

## Figures and Tables

**Figure 1 pharmaceuticals-14-00112-f001:**
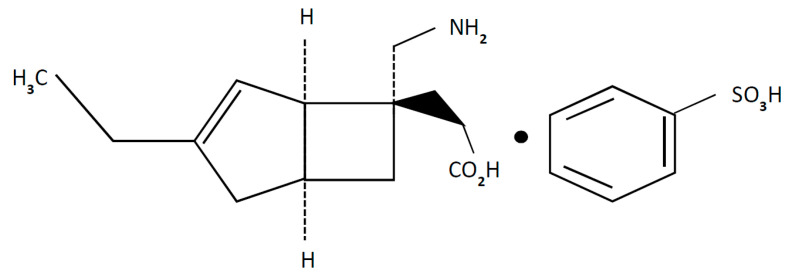
Chemical structure of mirogabalin besylate.

**Figure 2 pharmaceuticals-14-00112-f002:**
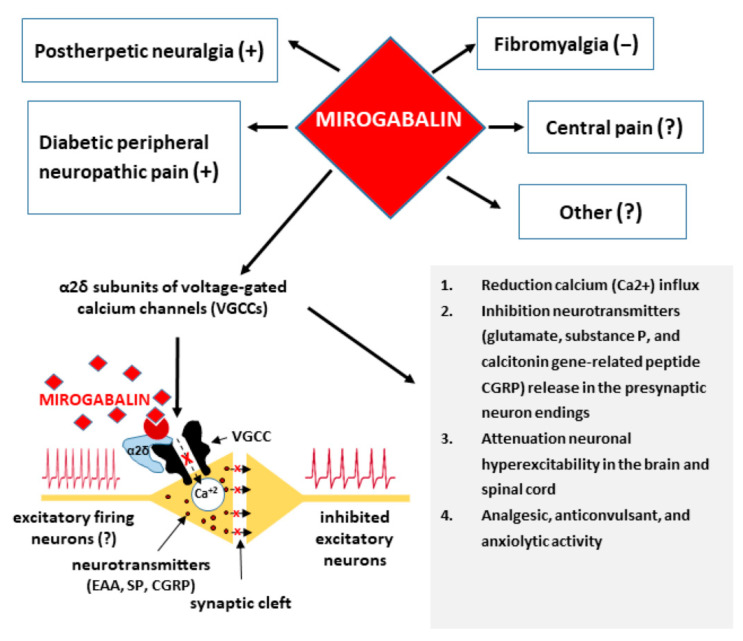
Mechanisms of action and therapeutic effects of mirogabalin.

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
