# Peer review of "Mirogabalin—A Novel Selective Ligand for the α2δ Calcium Channel Subunit"

_pharmaceuticals, 2021, doi:10.3390/ph14020112_

Round 1
Reviewer 1 Report
The review article “Mirogabalin-a novel selective ligand for the α2δ calcium channel subunit” by Zajaczkowska et al is a review article which describes the efficacy of mirogabalin on neuropathic pain including diabetic neuropathy and postherpetic neuralgia. The authors discuss present data on its pharmacodynamics and pharmacokinetics and also discuss reviews regarding experimental and clinical studies. In addition, the authors discuss not only the efficacy and safety but also the side effects of mirogabalin. This topic is of importance for treatment of chronic pain. The manuscript has been well written. However, I have some concerns on the paper.
Please add the review about which subtype of calcium channels (N, P, Q, L and T?) are involved in the effect of mirogabalin!
Lines 68-71: please add references at the end of this sentence.
Lines 117-118: please add references at the end of this sentence.
Lines 119-122: please add references at the end of this sentence.
Author Response
Thank you, we are very grateful for your thoughtful suggestions. Based on these remarks, we have made careful modifications to the original manuscript. The changes in the manuscript are highlighted in yellow.
The English was corrected by American Journal Experts (verification code 617F-6647-4DCF-464C-E928), however taking into account Referee comments it was carefully verified by us.
According to the Referee suggestion we have added new fragment showing that mirogabalin inhibits N-type calcium channel currents in rat DRG culture neurons. Moreover, we have added all the missing references.
Reviewer 2 Report
Overall, the review addresses most aspects of the topic on mirogabalin well. However, there are grammatical errors and long-winded sentences throughout. These need to be edited to improve readability.
I have more specific points below:
- Line 58, what does PNP stand for? The authors use DPNP in line 56, but not PNP. Please clarify.
- There needs to be a lot more discussion of the proposed cellular actions of mirogabalin based on what is known of alpha2delta1 and its role in pain pathways. For example, the authors have not cited work on the trafficking of alpha2delta1 (PMID: 30487217, 28256585, 25878262, 20861389, 20298215, 19339603).
- The authors should discuss other mechanisms of action of mirogabalin e.g. effects on anxiety circuitry and behavior (PMID: 32270470, 31982462).
- Figure 2: Is there a published study on effects of mirogabalin excitatory v inhibitory neurons? The figure assumes that mirogabalin preferentially acts on excitatory neurons, but this appears to be speculative. Please edit this to be clear whether this is known or unknown (insert question mark). This should also be included in the discussion of the proposed cellular actions of mirogabalin (mentioned in the previous point).
- Figure 2: The trace shown for “inhibited excitatory neurons” is confusing since it depicts smaller action potentials. Please show fewer APs as opposed to smaller APs.
References:
Nieto-Rostro M, Ramgoolam K, Pratt WS, Kulik A, Dolphin AC. Ablation of α2δ-1 inhibits cell-surface trafficking of endogenous N-type calcium channels in the pain pathway in vivo. Proc Natl Acad Sci U S A. 2018 Dec 18;115(51):E12043-E12052. doi: 10.1073/pnas.1811212115. Epub 2018 Nov 28. PMID: 30487217; PMCID: PMC6305000.
Kadurin I, Rothwell SW, Lana B, Nieto-Rostro M, Dolphin AC. LRP1 influences trafficking of N-type calcium channels via interaction with the auxiliary α2δ-1 subunit. Sci Rep. 2017 Mar 3;7:43802. doi: 10.1038/srep43802. PMID: 28256585; PMCID: PMC5335561.
D'Arco M, Margas W, Cassidy JS, Dolphin AC. The upregulation of α2δ-1 subunit modulates activity-dependent Ca2+ signals in sensory neurons. J Neurosci. 2015 Apr 15;35(15):5891-903. doi: 10.1523/JNEUROSCI.3997-14.2015. PMID: 25878262; PMCID: PMC4397591.
Tran-Van-Minh A, Dolphin AC. The alpha2delta ligand gabapentin inhibits the Rab11-dependent recycling of the calcium channel subunit alpha2delta-2. J Neurosci. 2010 Sep 22;30(38):12856-67. doi: 10.1523/JNEUROSCI.2700-10.2010. PMID: 20861389; PMCID: PMC6633565.
Bauer CS, Rahman W, Tran-van-Minh A, Lujan R, Dickenson AH, Dolphin AC. The anti-allodynic alpha(2)delta ligand pregabalin inhibits the trafficking of the calcium channel alpha(2)delta-1 subunit to presynaptic terminals in vivo. Biochem Soc Trans. 2010 Apr;38(2):525-8. doi: 10.1042/BST0380525. PMID: 20298215.
Bauer CS, Nieto-Rostro M, Rahman W, Tran-Van-Minh A, Ferron L, Douglas L, Kadurin I, Sri Ranjan Y, Fernandez-Alacid L, Millar NS, Dickenson AH, Lujan R, Dolphin AC. The increased trafficking of the calcium channel subunit alpha2delta-1 to presynaptic terminals in neuropathic pain is inhibited by the alpha2delta ligand pregabalin. J Neurosci. 2009 Apr 1;29(13):4076-88. doi: 10.1523/JNEUROSCI.0356-09.2009. PMID: 19339603; PMCID: PMC6665374.
Murasawa H, Kobayashi H, Yasuda SI, Saeki K, Domon Y, Arakawa N, Kubota K, Kitano Y. Anxiolytic-like effects of mirogabalin, a novel ligand for α2δ ligand of voltage-gated calcium channels, in rats repeatedly injected with acidic saline intramuscularly, as an experimental model of fibromyalgia. Pharmacol Rep. 2020 Jun;72(3):571-579. doi: 10.1007/s43440-020-00103-4. Epub 2020 Apr 8. PMID: 32270470.
Iwai T, Kikuchi A, Oyama M, Watanabe S, Tanabe M. Mirogabalin prevents repeated restraint stress-induced dysfunction in mice. Behav Brain Res. 2020 Apr 6;383:112506. doi: 10.1016/j.bbr.2020.112506. Epub 2020 Jan 23. PMID: 31982462.
Author Response
Thank you, we are very grateful for your thoughtful suggestions. Based on these remarks, we have made careful modifications to the original manuscript. The changes in the manuscript are highlighted in yellow.
The English was corrected by American Journal Experts (verification code 617F-6647-4DCF-464C-E928), however taking into account Referee comments it was carefully verified by us.
Ad. 1. According to the Referee suggestion we have corrected the sentence on line 58.
Ad. 2. According to the Referee suggestion we have added new fragments and references (PMID: 30487217, 28256585, 25878262, 20861389, 20298215, 19339603) into the chapter 4.
Ad. 3. According to the Referee suggestion we have added new fragments and references (PMID: 32270470, 31982462) into the chapter 6.1.
Ad. 4. According to the Referee suggestion we have introduced changes on Fig. 2 and into the text – marked in yellow.
Ad. 5. According to the Referee suggestion we have corrected Fig. 2
Round 2
Reviewer 1 Report
The authors addressed all my concerns.
Author Response
Thank you for your effort to review your paper and your valuable suggestions.